# Plasma and urine metabolomic analyses in aortic valve stenosis reveal shared and biofluid-specific changes in metabolite levels

**Cynthia Al Hageh[1,2], Ryan Rahy[2], Georges Khazen[2], Francois Brial[1], Rony S. Khnayzer[2]**\*, **Dominique Gauguier[1,3]**\*, **Pierre A. Zalloua[4]**\*

**1** Université de Paris, INSERM UMRS 1124, Paris, France, **2** Department of Natural Sciences, School of Arts and Sciences, Lebanese American University, Beirut, Lebanon, **3** McGill University and Genome Quebec Innovation Centre, Montreal, QC, Canada, **4** School of Medicine, University of Balamand, Amioun, Lebanon

☯ These authors contributed equally to this work.
\* dominique.gauguier@inserm.fr (DG); rony.khnayzer@lau.edu.lb (RSK); pierre.zalloua@balamand.edu.lb (PAZ)

## Abstract

Aortic valve stenosis (AVS) is a prevalent condition among the elderly population that eventually requires aortic valve replacement. The lack of reliable biomarkers for AVS poses a challenge for its early diagnosis and the application of preventive measures. Untargeted gas chromatography mass spectrometry (GC-MS) metabolomics was applied in 46 AVS cases and 46 controls to identify plasma and urine metabolites underlying AVS risk. Multivariate data analyses were performed on pre-processed data (e.g. spectral peak alignment), in order to detect changes in metabolite levels in AVS patients and to evaluate their performance in group separation and sensitivity of AVS prediction, followed by regression analyses to test for their association with AVS. Through untargeted analysis of 190 urine and 130 plasma features that could be detected and quantified in the GC-MS spectra, we identified contrasting levels of 22 urine and 21 plasma features between AVS patients and control subjects. Following metabolite assignment, we observed significant changes in the concentration of known metabolites in urine (n = 14) and plasma (n = 15) that distinguish the metabolomic profiles of AVS patients from healthy controls. Associations with AVS were replicated in both plasma and urine for about half of these metabolites. Among these, 2-Oxovaleric acid, elaidic acid, myristic acid, palmitic acid, estrone, myo-inositol showed contrasting trends of regulation in the two biofluids. Only trans-Aconitic acid and 2,4-Di-tert-butylphenol showed consistent patterns of regulation in both plasma and urine. These results illustrate the power of metabolomics in identifying potential disease-associated biomarkers and provide a foundation for further studies towards early diagnostic applications in severe heart conditions that may prevent surgery in the elderly.

**Data Availability Statement:** The data underlying the results presented in the study are available in S4 and S5 Tables.

**Funding:** C Al Hageg is funded by a PhD studentship of the Lebanese National Center for Scientific Research. R Khnayzer acknowledges support from the School Research and Development Council at the Lebanese American University (srdc-r-2017-20). The patient cohort was collected by P Zalloua with the financial support from the European Commission (FGENTCARD, LSHGCT-2006-037683). The funders had no role in study design, data collection and analysis, decision to publish, or preparation of the manuscript.

**Competing interests:** The authors have declared that no competing interests exist.

## Introduction

Aortic valve stenosis (AVS) results from inflammation caused by mechanical stress, lipid infiltration leading to fibrosis, leaflet thickening, and eventually calcification [1, 2]. The risk of AVS increases with age with 10% of AVS patients being above the age of 80 [3]. Pharmacologic treatments of AVS are often ineffective [4] and surgical valve repair or replacement is eventually needed in the elderly when surgery is often problematic [5]. Elevated concentration of plasma lipopotein(a) remains the most robust marker for AVS that may account for disease pathophysiology and valve molecular anomalies described in AVS [6]. Nevertheless, early diagnosis and prognosis of AVS can be greatly improved by the high-throughput measurement of reliable molecular biomarkers in easily accessible biospecimens.

Metabolomics provides a platform for biomedical discovery, as well as clinical and pharmaceutical applications, which has been extensively used for biomarker discovery, drug response ascertainment and disease pathway identification [7]. It relies on the qualitative and quantitative analysis of small molecular weight metabolites, which are end products of genome expression while integrating consequences of environmental exposures [8]. It has been successfully used for in depth characterisation of metabolic changes in health and disease [9] and particularly powerful to identify metabolites associated with increased risk of metabolic and vascular diseases [10–12]. The application of metabolomics to test associations between AVS and many metabolites simultaneously represents a prospect of significant advances for early disease diagnosis and improved treatment efficacy.

Here, we applied highly sensitive untargeted metabolomics based on gas chromatography mass spectrometry to identify metabolites associated with AVS. Through plasma and urine paralleled metabolomic profiling of AVS patients and control subjects, we sought to identify a series of metabolic features associated with the disease. We also investigated the existence of metabolites showing either shared or biofluid-specific association with AVS. These results underline the power of metabolomics to identify potential biomarkers for early AVS diagnosis and targets for therapeutic applications that may prevent or anticipate the need for cardiac surgery in the elderly.

## Material and methods

### Study subjects and AVS diagnosis

Subjects were recruited as part of a comprehensive study on coronary artery disease between 2007 and 2009 [13]. They were selected based on the presence (cases) or absence (controls) of AVS as clinically determined by the occurrence of a systolic murmur in the aortic valve area, which was subsequently confirmed in cases by echocardiography. Urine and plasma samples from 46 AVS patients and 46 healthy controls (matched for sex and age ± 5 years) were used in this study. About 30 ml of urine and 20 ml of arterial blood were collected in subjects after 12 hours fasting. Blood was collected on EDTA and plasma was separated by centrifugation at room temperature. Urine samples were centrifuged at room temperature. Plasma and urine aliquots were stored at -80˚C until metabolomic analysis.

All subjects provided a written informed consent, and the study protocol was approved by the International Review Board (IRB) at the Lebanese American University. All protocols were performed according to the Helsinki Declaration of 1975.

### Gas chromatography coupled with mass spectrometry

Samples were prepared for metabolite extraction using methods optimized for urine [14] and plasma [15]. The internal standard 2-isopropylmalic acid was used for quality control.

Trimethylsilylation was applied in sample preparation of urine and plasma extracts for gas chromatography mass spectrometry (GC-MS) acquisition. Samples were subjected to GC-MS HP6890 (Agilent Technologies, Santa Clara, CA) equipped with a capillary column HP-5MS 5% phenyl methyl siloxane of 30m nominal length, 250μm nominal diameter and 0.25μm nominal film thickness (Agilent Technologies, Santa Clara, CA). A 1μL aliquot of the derivatized solution was injected under split mode with a ratio 3:1 using Helium gas. GC-MS raw chromatograms were exported in CDF format for data pre-processing, and CSV files were obtained which included peak retention time, peak height, peak area and metabolites identification using the NIST08 library (https://chemdata.nist.gov/). Metabolite annotation was manually checked using a similarity criterion of ≥80%. Data from negative controls (same reagents and conditions excluding sample) were acquired with GC-MS in order to remove artifact peaks from the solvents used in extraction and derivatization.

## Metabolomic data pre-processing

GC-MS raw data were pre-processed to generate a comprehensive peak table that included all detected peaks characterized by a specific retention time (RT), mass to charge ratio (m/z) and the intensity of each peak across multiple samples from multiple sample groups. The XCMS (v 3.6.1) tool in R statistical language (through Bioconductor v 1.30.4) was used for GC-MS data pre-processing where RT were aligned, and signal drift and batch effect were corrected. XCMS uses CDF format as input, and gives a data Matrix table as output. Using XCMS, peak detection was performed while the peak width parameter was set visually after assessing the chromatographic peaks belonging to the internal standard. Thus, based on the internal standard, the range of RT values was set between 450 and 460 seconds, the m/z range was between 275 and 278 and the value of maximum expected deviation of m/z values was set to 3ppm. Then, peak alignment was performed so that all RTs can be adjusted to match across all samples.

## Metabolomic data processing

The MetaboAnalyst tool (v 2.0.1) in R package was used for statistical analysis. Each spectral feature was normalized to the internal standard 2-isopropylmalic acid. In a separate analysis, data generated from urine samples were normalized to creatinine, which is proposed as an alternative method to internal standard to account for urine dilution [16]. A generalized logarithm transformation was subsequently applied for data transformation. Then univariate analysis using volcano plot and multivariate analysis using principle component analysis (PCA), Partial Least Squares-Discriminant Analysis (PLS-DA) and Orthogonal PLS-DA (OPLS-DA) [17] were performed. Volcano plot showed the combination between the fold-change (log 2 (FC)) of the relative abundance of each spectral feature in AVS cases and controls and the statistical significance of the FC. Model cross validation with $R^2$ and $Q^2$ was used to assess the goodness of fit and predictability of the OPLS-DA model respectively. The index of Variable Importance in Projection (VIP), which measures the importance of individual metabolite features in the PLS-DA model, was used to weigh their contribution to the separation between cases and controls. Since OPLS-DA tends to over fit data a permutation test with 1,000 iterations was performed to validate the model and understand the significance of class discrimination. To decrease the rate of false positives in the selection, q-values were calculated using Benjamini-Hochberg method [18] and the threshold was set at 0.05. Metabolites were selected as candidates when VIP>1, False Discovery Rate (FDR) <0.05 and q-values <0.05. Receiver operating characteristic (ROC) analysis was developed using the Biomarker Analysis tool in MetaboAnalyst (www.metaboanalyst.ca) to evaluate the performance of each candidate metabolite to separate cases and controls. Area under the ROC curve (AUC) for each metabolite, as

well as their 95% confidence intervals, were used to assess the utility of the candidate metabo-
lites according to criteria [19] designed to rank candidate biomarkers as excellent (AUC = 0.9–
1.0), good (AUC = 0.8–0.9), fair (AUC = 0.7–0.8), poor (AUC = 0.6–0.7) or failed
(AUC = 0.5–0.6).

Generalized linear models (GLMs) were used to determine the metabolomic peaks signifi-
cantly associated with AVS. After adjusting for age, sex, body mass index (BMI), hyperlipid-
emia and diabetes, logistic regression was used to assess the association of the metabolite peaks
with AVS. The p-values obtained corresponded to the p-value of the peak in each model.
These values were then corrected using the Benjamini-Hochberg method [18]. Peaks were
considered to be statistically significant when their adjusted p-values (q-values) were less than
0.05.

### Biological pathway analysis

Analysis of the biological pathways in the Kyoto Encyclopedia of Genes and Genomes (KEGG,
www.genome.jp/kegg) underlying AVS risk was carried out with data from urine and plasma
metabolites that were significantly associated with AVS using the web-tool MetaboAnalyst
(www.metaboanalyst.ca). Both over-representation of significantly altered metabolites within
pathways (P-values based on hypergeometric test) and the impact of metabolite changes on
the function of the pathway through alterations in critical junction points of the pathway (rela-
tive betweenness centrality) were assessed.

## Results

### Clinical and biochemical features of AVS patients and control individuals

The 92 subjects were phenotypically well characterized with a mean age of 59.1 (±1.3) years, a
mean body weight of 81.9 kg (±1.7), a mean BMI of 30.8 kg/$m^2$ (±0.5), a mean blood glucose
112.3 mg/dL (±5.3), a mean triglyceride of 186.5 mg/dL (±9.0), a mean HDL-cholesterol of
40.1 mg/dL (±1.4), a mean of LDL-cholesterol of 115.0 mg/dL (±4.3) and mean of total choles-
terol of 188.3 mg/dL (±5.1) (**Table 1**). A total of 63 subjects were hypertensive (68.5%), 68 had
family history of hypertension (73.9%), 18 were diagnosed with diabetes (19.6%), 40 were
hyperlipidemic (43.5%), 52 had family history of diabetes (56.5%) and 37 had family history of
hyperlipidemia (40.2%). There were no significant differences between AVS patients and con-
trol individuals for biochemical variables. Markedly reduced serum LDL-cholesterol in AVS
subjects when compared to controls was not statistically significant (p = 0.215). There were no
significant differences between males and females in any of these variables (**S1 Table**).

### General features of metabolomic profiling data

Using a signal to noise ratio of 6 applied to peak detection on the GC-MS chromatograms, a
total of 190 and 130 peaks have been confidently detected in urine and plasma, respectively.
Using the NIST08 library, a total of 112 and 70 metabolites possessing a similarity
index ≥ 80% were detected with GC-MS in urine and plasma extracts respectively (**S2 Table**).
The intensity of each peak was measured and normalized to the internal standard (2-isopro-
pylmalic acid).

### Metabolomic analysis of urine and plasma samples in AVS patients

Using volcano plots, we identified 30 features showing evidence of difference in urine levels
(nominal p<0.05) between AVS patients and controls, including 21 features which were more
abundant in AVS patients than in controls (**Fig 1A; S3 Table**). In the PCA score plots derived

**Table 1. Clinical and biochemical features of the 92 subjects selected for presence or absence of aortic valve stenosis.**

|  | Total (92) | Controls (46) | Cases (46) | p-value |
|---|---|---|---|---|
| **Age** | 59.1 ± 1.3 | 59.5 ± 1.9 | 58.8 ± 1.9 | 0.81 |
| **Body weight (Kg)** | 81.9 ± 1.7 | 82.2 ± 2.4 | 81.6 ± 2.3 | 0.86 |
| **Body mass index (Kg/m$^2$)** | 30.8 ± 0.5 | 30.9 ± 0.8 | 30.8 ± 0.7 | 0.93 |
| **Plasma glucose (mg/dL)** | 112.3 ± 5.3 (49) | 112.2 ± 6.6 (36) | 112.6 ± 7.5 (13) | 0.97 |
| **Triglycerides (mg/dL)** | 186.5 ± 9.0 (80) | 186.6 ± 13.2 (42) | 186.4 ± 12.2 (38) | 0.99 |
| **HDL cholesterol (mg/dL)** | 40.1 ± 1.4 (82) | 40.1 ± 1.9 (43) | 40.0 ± 2.1 (39) | 0.98 |
| **LDL cholesterol (mg/dL)** | 115.0 ± 4.3 (80) | 120.0 ± 6.2 (42) | 109.4 ± 5.7 (38) | 0.22 |
| **Total cholesterol (mg/dL)** | 188.3 ± 5.1 (82) | 191.0 ± 8.0 (43) | 185.3 ± 6.3 (39) | 0.58 |
| **Diagnosed diabetic (%)** | 18 (19.6%) | 7 (15.2%) | 11 (23.9%) |  |
| **Diagnosed hypertensive (%)** | 63 (68.5%) | 27 (58.7%) | 36 (78.3%) |  |
| **Diagnosed hyperlipidemic (%)** | 40 (43.5%) | 12 (26.1%) | 28 (60.9%) |  |
| **FH diabetes (%)** | 52 (56.5%) | 29 (63.0%) | 23 (50.0%) |  |
| **FH hypertension (%)** | 68 (73.9%) | 34 (73.9%) | 34 (73.9%) |  |
| **FH hyperlipidemia (%)** | 37 (40.2%) | 16 (34.8%) | 21 (45.7%) |  |

FH, Family History. Data are means ± SEM.

from GC-MS spectra, a clear separation was obtained to differentiate metabolome profiles of AVS patients and controls (**Fig 1B**). The principal components PC1, PC2, and PC3 described 26.8%, 16.6% and 9.2% of the variation, respectively. The OPLS-DA score plot also provided a clear separation of AVS patients and controls (**Fig 1C**). The goodness of fit values of the OPLS-DA model were 0.113 ($R^2X$) and 0.76 ($R^2Y$), with a predictive ability value of 0.735 ($Q^2$) (**S1A Fig**). This model explained 11.3% of the variation in metabolites levels and 76.0% of the variation between the groups, and the average prediction capability was 73.5%. The difference between $R^2Y$ and $Q^2$ was less than 0.2 and the $Q^2$ value was greater than 50%, revealing an

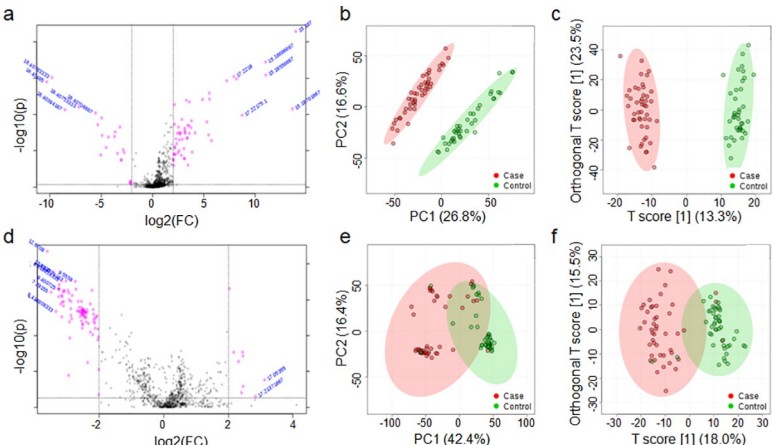

**Fig 1. Discrimination analysis of AVS patients and control individuals through metabolomic profiling of biofluids.** Metabolomic data were derived from GC-MS spectra of urine (a-c) and plasma (d-f) samples from AVS patients (n = 46) and healthy controls (n = 46). Univariate analysis of GC-MS spectral data in the 92 samples was performed to identify metabolomic features significantly separating cases and controls (nominal p<0.05), which are plotted in pink dots in the upper part of the volcano plots (a, d). Multivariate statistics were applied to perform principle component analysis (PCA) (b, e) and orthogonal partial least squares-discriminant analysis (OPLS-DA) (c, f) and assess sample classification in cases and controls. The 95% confidence regions are displayed by shaded ellipses in AVS patients (red) and controls (green).

excellent predictive capability. Permutation tests were performed (1000 iterations) to verify that this OPLS-DA model was not random or due to over fitting (p<0.001) (**S1B Fig**).

To improve the metabolic profile of AVS patients, we complemented urine metabolic analyses with GC-MS metabolomic profiling of plasma samples from the same panel of AVS and control subjects. As previously observed with urine metabolic features, volcano plot analysis identified 23 significantly contributing features (nominal p<0.05), including 10 features that were more abundant in AVS patients than in controls (**Fig 1D**; **S3 Table**). The PCA score plots of plasma metabolomics provided some evidence of clustering of AVS and control groups that was however inferior to that achieved with urine data (**Fig 1E**). The PC1, PC2 and PC3 described 42.4%, 16.4%, and 7.7% of the variation, respectively. The OPLS-DA plot also indicated that the two groups are well separated into 2 clusters (**Fig 1F**). OPLS-DA showed significantly good predictability ($Q^2 = 0.649$), and good capability to explain the metabolic variation between AVS patients and controls ($R^2Y = 0.684$), with goodness of fit values of 0.180 ($R^2X$) and 0.684 ($R^2Y$) (**S1C Fig**). The model explained 18.0% of the variation in metabolite levels and 68.4% of the variation between the groups, and a higher average prediction capability (94.9%) than urine data (73.5%). The difference between $R^2Y$ and $Q^2$ (<0.2) and the $Q^2$ value (>50%) confirmed the excellent predictive capability of the model (0.684). The permutation test indicated that AVS had significant impacts on the plasma metabolic profiling (p<0.001; 1,000 iterations) (**S1D Fig**).

## Urine and plasma metabolomic profiling in AVS underlines biofluid-specific changes in metabolite abundance

We used the index of Variable Importance in Projection (VIP) derived from the PLS-DA models of urine and plasma metabolomic datasets to weigh the impact of each individual metabolite feature to separate AVS cases and controls (**Fig 2**). Following feature annotations using the NIST08 library, we identified a total of 16 known urine metabolites significantly contributing to the separation between AVS and controls (VIP>1, nominal p<0.05, q<0.05). These include

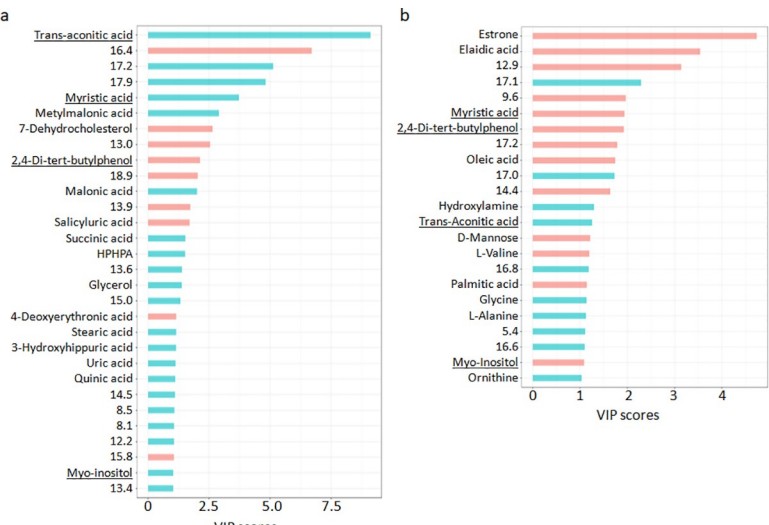

**Fig 2. Contribution of metabolites in the separation of AVS cases and controls.** The Variable Importance in Projection (VIP) was used to weigh the contribution of urine (a) and plasma (b) metabolomic features to the separation between cases and controls in the PLS-DA model. Data were normalized to the internal standard 2-isopropylmalic acid. Upregulated (blue bars) and downregulated (red bars) features are shown. Metabolites found associated with AVS in both urine and plasma are underlined. Details of metabolite features are given in **S3 Table**.

trans-Aconitic acid, myristic acid, methylmalonic acid, 7-Dehydrocholesterol, 2,4-Di-tert-butylphenol, malonic acid, 2-Hydroxyhippuric acid, 3-Hydroxyhippuric acid, succinic acid, glycerol, quinic acid, uric acid, stearic acid, 4-Deoxyerythronic acid, 3-(3-Hydroxyphenyl)-3-Hydroxypropanoic acid (HPHPA) and myo-inositol (**Fig 2A**, **S3 Table**). We included elaidic acid in the list of annotated metabolites even though it is not reported as yet in human urine in the Human Metabolome Database (HMDB).

Results from association analysis to AVS were generally conserved when urine metabolomic data were normalized to creatinine (**S4 Table**). The metabolites salicyluric acid (2-Hydroxyhippuric acid), myo-inositol, glycerol, 4-Deoxyerythronic acid, uric acid and two unknown metabolites at RT 8.1 and 15.8mins were associated to AVS only following normalization to the internal standard. On the other hand, associations between AVS and the metabolites p-Hydroxyphenylacetic acid, palmitic acid, oxoadipic acid, hypoxanthine, estrone, D-Glucose and two unknown metabolites at RT 13.3 and 13.5mins were significant only when data were normalized to creatinine. The remaining metabolites consistently associated to AVS using the two normalization procedures showed similar VIP, similar magnitude and direction of changes between AVS and controls and consistent magnitude statistical significance of association. Strong conservation of our results derived through two different normalization methods underlines the robustness of our findings.

Among the plasma metabolic features significantly contributing to the separation between AVS and controls, 14 could be attributed to known metabolites (elaidic acid, palmitic acid, oleic acid, myristic acid, trans-Aconitic acid, D-Mannose, estrone, L-Alanine, L-Valine, myo-inositol, ornithine, hydroxylamine, 2,4-Di-tert-butylphenol, glycine) (**Fig 2B**, **S3 Table**). According to HMDB, 2,4-Di-tert-butylphenol has not been previously reported in human plasma.

In the urine dataset normalized to the internal standard, levels of myristic acid, trans-Aconitic acid, myo-inositol and 2,4-Di-tert-butylphenol were different between AVS patients and controls in both plasma and urine, but myristic acid and myo-inositol showed discordant direction of changes in the two biofluids. In addition when urine metabolomic data were normalized to creatinine, estrone and palmitic acid also showed opposite direction of changes in AVS in plasma and urine.

## Biofluid metabolomic profiling data suggest novel candidate metabolite biomarkers for AVS

The performance of each urine and plasma candidate metabolite to separate AVS cases and controls was evaluated by ROC curve analysis (**S2** and **S3** **Figs**). AUC values with their p-values and FC for each urine and plasma metabolite associated with AVS are summarized in **S3 Table**. The majority of potential metabolite biomarkers showed good to excellent (AUC>0.8) discriminant capability. In urine, trans-Aconitic acid, myristic acid, methylmalonic acid, 7-Dehydrocholesterol, 2,4-Di-tert-butylphenol, succinic acid, malonic acid were excellent potential biomarkers (AUC = 0.90–1.00). The known metabolites 3-(3-Hydroxyphenyl)-3-Hydroxypropanoic acid (HPHPA), quinic acid, 4-Deoxyerythronic acid and uric acid were good potential biomarkers (AUC = 0.80–0.90). The remaining candidates (2-Hydroxyhippuric acid, 3-Hydroxyhippuric acid, stearic acid, glycerol, and myo-inositol) were fair biomarkers (AUC = 0.7–0.8) (**S3 Table**).

Excellent potential plasma biomarkers showing AUC above 0.90 include elaidic acid, estrone, palmitic acid, myristic acid, 2,4-Di-tert-butylphenol, oleic acid and myo-inositol. Glycine, hydroxylamine, trans-Aconitic acid, L-Alanine and L-Valine were good biomarkers (AUC = 0.80–0.90), whereas ornithine and D-Mannose were not considered as good biomarkers (**S3 Table**).

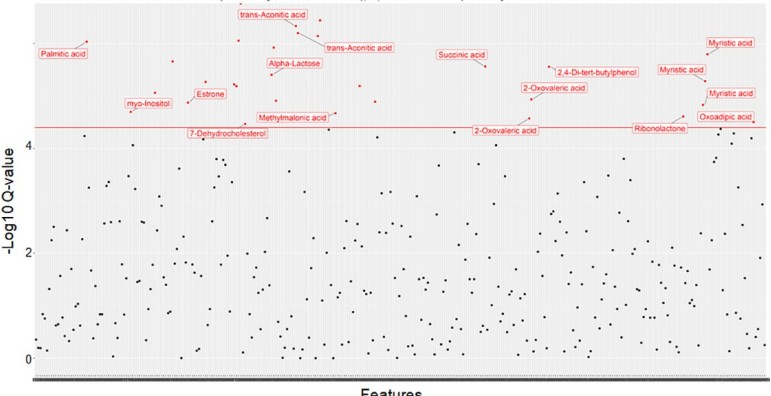

**Fig 3. Association analysis of urine GC-MS spectral data in AVS patients and control individuals.** Data were derived by GC-MS analysis of urine samples from 46 patients and 46 controls. Generalized linear models were used to determine significant associations between metabolomic peaks and AVS after adjusting for age, sex, body mass index, hyperlipidemia and diabetes, and correcting for multiple testing. Signal intensities normalized to the internal standard 2-isopropylmalic acid are plotted against the Q-values. Features showing evidence of statistically significant association with AVS (q values<0.05) are shown with red dots.

These analyses also pointed to GC-MS signals associated with AVS that correspond to unknown metabolites in urine (n = 9) and plasma (n = 9) and show fair to excellent capacity to separate AVS patients to control individuals.

## Statistical association of urine and plasma GC-MS features identifies metabolites underlying AVS risk

Following adjustment for age, sex, BMI, hyperlipidemia and diabetes, we identified statistically significant associations between AVS and 30 features in urine (**Fig 3**) and 35 features in plasma (**Fig 4**). In several instances, several independent features were attributed to a single metabolite. Among these features, 14 urine metabolites and 15 plasma metabolites could be identified based on available information in the NIST08 data repository (**Tables 2 and 3**). Urine

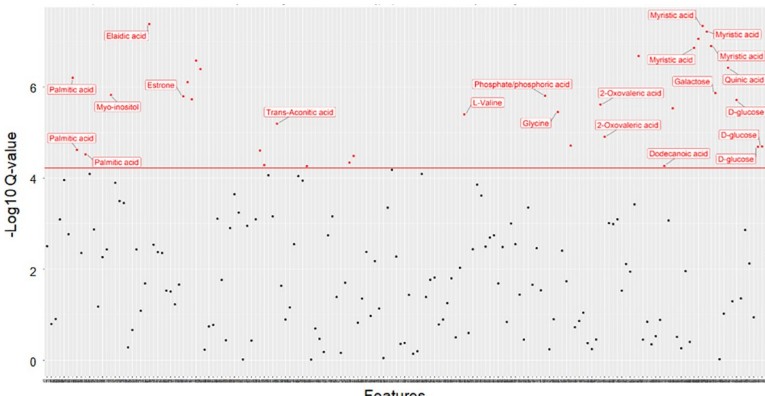

**Fig 4. Association analysis of plasma GC-MS spectral data in AVS patients and control individuals.** Data were derived by GC-MS analysis of plasma samples from 46 patients and 46 controls. Generalized linear models were used to determine significant associations between metabolomic peaks and AVS after adjusting for age, sex, body mass index, hyperlipidemia and diabetes, and correcting for multiple testing. Signal intensities normalized to the internal standard 2-isopropylmalic acid are plotted against the Q-values. Features showing evidence of statistically significant association with AVS (q values<0.05) are shown with red dots.

**Table 2. Urinary metabolites contributing to the separation between the AVS patients and healthy controls.**

| Metabolite | RT | Normalization internal standard | | | | Normalization creatinine | | | | Regulation in AVS |
|---|---|---|---|---|---|---|---|---|---|---|
| | | q-value | RC | CI 2.5 | CI 97.5 | q-value | RC | CI 2.5 | CI 97.5 | |
| Methylmalonic acid | 2.7 | 0.0301 | 4.78 | 2.78 | 7.24 | NS | - | - | - | Positive |
| Unknown | 3.4 | 0.0183 | 4.98 | 3.10 | 7.66 | 0.0099 | 9.95 | 6.25 | 15.1 | Positive |
| Unknown | 3.5 | NS | - | - | - | 0.0291 | 5.41 | 3.20 | 8.24 | Positive |
| Succinic acid | 5.5 | 0.0040 | 4.82 | 3.02 | 7.10 | 0.0273 | 6.88 | 4.06 | 10.44 | Positive |
| 4-Deoxyerythronic acid | 6.3 | NS | - | - | - | 0.0384 | 4.15 | 2.41 | 6.35 | Positive |
| 2-Oxovaleric acid | 6.8 | 0.0376 | 3.83 | 2.22 | 5.82 | 0.0131 | 7.50 | 4.51 | 11.21 | Positive |
| Unknown | 7.1 | NS | - | - | - | 0.0382 | 6.94 | 4.05 | 10.63 | Positive |
| 2,4-Di-tert-butylphenol | 7.3 | 0.0041 | -2.15 | -3.27 | -1.41 | 0.0009 | -5.52 | -8.02 | -3.61 | Negative |
| Erythronic acid | 7.4 | NS | - | - | - | 0.0372 | 3.29 | 1.90 | 5.02 | Positive |
| Unknown | 7.9 | NS | - | - | - | 0.0388 | 4.07 | 2.36 | 6.23 | Positive |
| 2-Deoxypentonic acid | 8.2 | NS | - | - | - | 0.0346 | 5.61 | 3.30 | 8.58 | Positive |
| D-Fructose | 8.3 | NS | - | - | - | 0.0342 | 5.07 | 2.98 | 7.75 | Positive |
| Unknown | 8.5 | NS | - | - | - | 0.0291 | 6.46 | 3.78 | 9.79 | Positive |
| Ribonolactone | 9.1 | 0.0343 | 1.40 | 0.82 | 2.13 | 0.0289 | 8.84 | 5.34 | 13.62 | Positive |
| Myristic acid | 9.4 | 0.0209 | 3.91 | 2.41 | 6.01 | 0.0111 | 4.69 | 2.97 | 7.16 | Positive |
| HPHPA | 9.5 | NS | - | - | - | 0.0153 | 5.84 | 3.58 | 8.88 | Positive |
| Quinic acid | 9.7 | NS | - | - | - | 0.008 | 3.94 | 2.40 | 5.83 | Positive |
| Oxoadipic acid | 9.84 | 0.0439 | 5.42 | 3.11 | 8.27 | 0.0468 | 7.37 | 4.22 | 11.28 | Positive |
| Palmitic acid | 10.6 | 0.0014 | 5.88 | 3.84 | 8.64 | 0.0048 | 8.07 | 5.04 | 11.93 | Positive |
| Myo-inositol | 11.1 | 0.0284 | 7.57 | 4.53 | 11.59 | NS | - | - | - | Positive |
| Elaidic acid | 11.7 | 0.0125 | 7.24 | 4.44 | 10.92 | 0.0251 | 7.36 | 4.33 | 11.13 | Positive |
| Stearic acid | 11.8 | NS | - | - | - | 0.048 | 8.32 | 4.82 | 12.80 | Positive |
| Unknown | 12.2 | 0.0032 | 3.24 | 2.07 | 4.80 | NS | - | - | - | Positive |
| D-Glucose | 12.4 | NS | - | - | - | 0.0186 | 4.86 | 2.93 | 7.37 | Positive |
| Unknown | 12.5 | NS | - | - | - | 0.0206 | 5.38 | 3.18 | 8.10 | Positive |
| Estrone | 12.7 | 0.0190 | 3.40 | 2.11 | 5.22 | 0.0234 | 8.55 | 5.13 | 13.03 | Positive |
| Unknown | 13 | 0.0079 | -1.58 | -2.44 | -1.02 | 0.0109 | -4.47 | -6.79 | -2.78 | Negative |
| Unknown | 13.3 | NS | - | - | - | 0.0057 | 4.83 | 2.98 | 7.13 | Positive |
| Unknown | 13.4 | NS | - | - | - | 0.031 | 4.29 | 2.54 | 6.57 | Positive |
| Unknown | 13.5 | 0.0087 | 1.74 | 1.05 | 2.58 | 0.01 | 6.10 | 3.72 | 9.12 | Positive |
| Unknown | 13.6 | 0.0094 | 5.02 | 3.23 | 7.75 | 0.006 | 6.75 | 4.29 | 10.17 | Positive |
| 7-Dehydrocholesterol | 14 | 0.0475 | -0.77 | -1.17 | -0.43 | NS | - | - | - | Negative |
| Unknown | 14.1 | NS | - | - | - | 0.031 | 5.89 | 3.50 | 9.02 | Positive |
| Alpha-Lactose | 14.4 | 0.0058 | 4.21 | 2.69 | 6.35 | 0.0224 | 11.95 | 7.44 | 18.58 | Positive |
| Unknown | 14.5 | 0.0018 | 4.16 | 2.67 | 6.08 | 0.0074 | 6.48 | 4.03 | 9.68 | Positive |
| Trans-Aconitic acid | 15.2 | 0.0007 | 1.57 | 1.05 | 2.30 | 0.0015 | 3.28 | 2.15 | 4.86 | Positive |
| Unknown | 16.4 | 0.0011 | -1.23 | -1.85 | -0.83 | 0.0012 | -4.11 | -6.13 | -2.74 | Negative |
| Unknown | 20.3 | 0.0094 | -2.00 | -2.98 | -1.22 | NS | - | - | - | Negative |

Data were derived by GC-MS analysis of urine samples from 46 patients and 46 controls. P-values were adjusted for age, sex, body mass index, hyperlipidemia and diabetes. Data are shown for urine metabolomic signals normalized to either the internal standard 2-isopropylmalic acid or creatinine. The regression coefficient (RC) illustrates the magnitude of the statistical effect on the increased or decreased concentration of the metabolites in AVS patients. RT, Retention Time; CI, Confidence Interval. Positive and negative regulation indicates up- and down-regulation of the metabolic features in AVS patients, respectively. HPHPA, 3-(3-Hydroxyphenyl)-3-hydroxypropanoic acid.

**Table 3. Plasma metabolites contributing to the separation between the AVS patients and healthy controls.**

| Metabolite | RT | q-value | Regression coefficient | CI 2.5 | CI 97.5 | Regulation in AVS |
|---|---|---|---|---|---|---|
| L-Valine | 4.4 | 0.0035 | -0.77 | -1.14 | -0.48 | Negative |
| Phosphate/phosphoric acid | 5.1 | 0.0014 | -1.81 | -2.64 | -1.14 | Negative |
| Glycine | 5.2 | 0.0031 | 0.91 | 0.56 | 1.34 | Positive |
| Unknown | 5.4 | 0.0167 | 2.08 | 1.27 | 3.21 | Positive |
| 2-Oxovaleric acid | 6.8 | 0.0021 | -1.86 | -2.74 | -1.17 | Negative |
| 2,4-Di-tert-butylphenol | 7.3 | 0.0002 | -0.98 | -1.40 | -0.65 | Negative |
| Dodecanoic acid | 8.0 | 0.0452 | -1.12 | -1.71 | -0.61 | Negative |
| Unknown | 8.4 | 0.0026 | -1.37 | -2.01 | -0.85 | Negative |
| Myristic acid | 9.4 | 0.0001 | -0.80 | -1.13 | -0.53 | Negative |
| Galactose | 9.5 | 0.0012 | -0.82 | -1.20 | -0.52 | Negative |
| Quinic acid | 9.7 | 0.0004 | -0.89 | -1.30 | -0.59 | Negative |
| D-Glucose | 9.9 | 0.0176 | 4.62 | 2.73 | 7.03 | Positive |
| Palmitic acid | 10.6 | 0.0006 | -1.71 | -2.50 | -1.13 | Negative |
| Myo-inositol | 11.1 | 0.0013 | -1.22 | -1.78 | -0.78 | Negative |
| Elaidic acid | 11.7 | <0.001 | -0.93 | -1.30 | -0.63 | Negative |
| Estrone | 12.7 | 0.0014 | -0.66 | -1.01 | -0.44 | Negative |
| Unknown | 12.9 | 0.0007 | -0.68 | -1.00 | -0.44 | Negative |
| Unknown | 14.5 | 0.0211 | -0.48 | -0.74 | -0.28 | Negative |
| Trans-Aconitic acid | 15.2 | 0.0056 | 0.90 | 0.55 | 1.34 | Positive |
| Unknown | 16.9 | 0.0455 | 0.63 | 0.35 | 0.97 | Positive |
| Unknown | 17.2 | 0.0382 | -0.33 | -0.50 | -0.18 | Negative |

Data were derived by GC-MS analysis of plasma samples from 46 patients and 46 controls. P-values were adjusted for age, sex, body mass index, hyperlipidemia and diabetes. The regression coefficient illustrates the magnitude of the statistical effect on the increased or decreased concentration of the metabolites in AVS patients. RT, Retention Time; CI, Confidence Interval. Positive and negative regulation indicates up- and down-regulation of the metabolic features in AVS patients, respectively.

metabolites significantly associated with AVS included myristic acid, palmitic acid, methylmalonic acid, succinic acid, 2-Oxovaleric acid, elaidic acid, ribonolactone, oxoadipic acid, myoinositol, estrone, α-lactose, trans-Aconitic acid, 7-Dehydrocholesterol and 2,4-Di-tert-butylphenol (**Table 2**). With the exception of methylmalonic acid, myo-inositol, 7-Dehydrocholesterol and two unknown metabolites at RT 12.2 and 20.3mins, statistically significant associations between AVS and urine metabolites were replicated when metabolomic data normalized to creatinine were used for statistical analysis (**Table 2**). Normalization of urine data to creatinine allowed the identification of several additional associations between AVS and metabolites, including 4-Deoxyerythronic acid, erythronic acid, 2-Deoxypentonic acid, D-Fructose, HPHPA, quinic acid, stearic acid, D-Glucose, 7-Dehydrocholesterol and nine unknown metabolites (**Table 2**).

We identified statistically significant associations between plasma features and AVS for glycine, D-Glucose, trans-Aconitic acid, myristic acid, palmitic acid, elaidic acid, L-Valine, 2,4-Di-tert-butylphenol, phosphoric acid, 2-Oxovaleric acid, dodecanoic acid, quinic acid, galactose, myo-inositol and estrone (**Table 3**).

AVS patients showed significantly elevated urinary concentrations of myristic acid, trans-Aconitic acid, methylmalonic acid, 2-Oxovaleric acid, oxoadipic acid, palmitic acid, elaidic acid, α-lactose, estrone, ribonolactone, succinic acid and myo-inositol (**Fig 5**, **Table 2**). By contrast, urine concentration of the remaining metabolites (2,4-Di-tert-butylphenol and 7-Dehydrocholesterol) were lower in AVS patients than in controls. Plasma concentration of the saturated fatty acids myristic acid and palmitic acid, dodecanoic acid, the unsaturated fatty

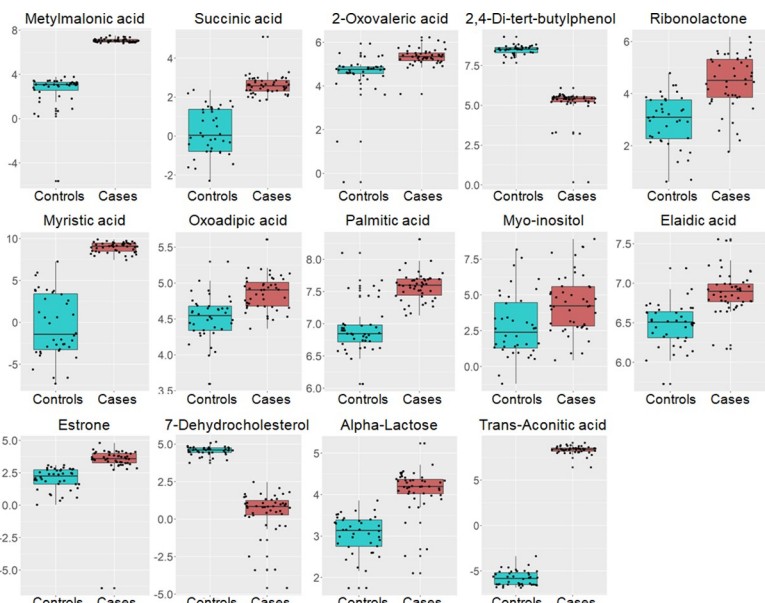

**Fig 5. Regulatory pattern of urine metabolites associated with AVS in patients and control individuals.** Data from urine candidate metabolites in the 46 AVS cases (orange boxes) and 46 controls (blue boxes) are shown. Data are normalized to the internal standard (2-isopropylmalic acid) and log transformed. The boxplots show the median and the inter-quartile range for each metabolite in the two groups.

acid elaidic acid, the essential amino acid L-Valine, estrone, phosphoric acid, 2-Oxovaleric acid, quinic acid, 2,4-Di-tert-butylphenol, myo-inositol and galactose were significantly lower in AVS patients than in controls (**Fig 6**, **Table 3**). In contrast, plasma concentration of the

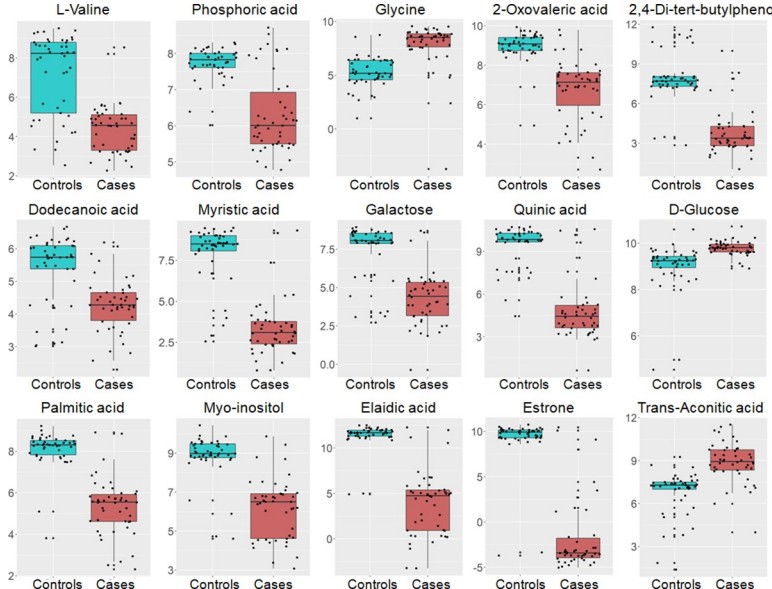

**Fig 6. Regulatory pattern of plasma metabolites associated with AVS in patients and control individuals.** Data from plasma candidate metabolites in the 46 AVS cases (orange boxes) and 46 controls (blue boxes) are shown. Data are normalized to the internal standard (2-isopropylmalic acid) and log transformed. The boxplots show the median and the inter-quartile range for each metabolite in the two groups.

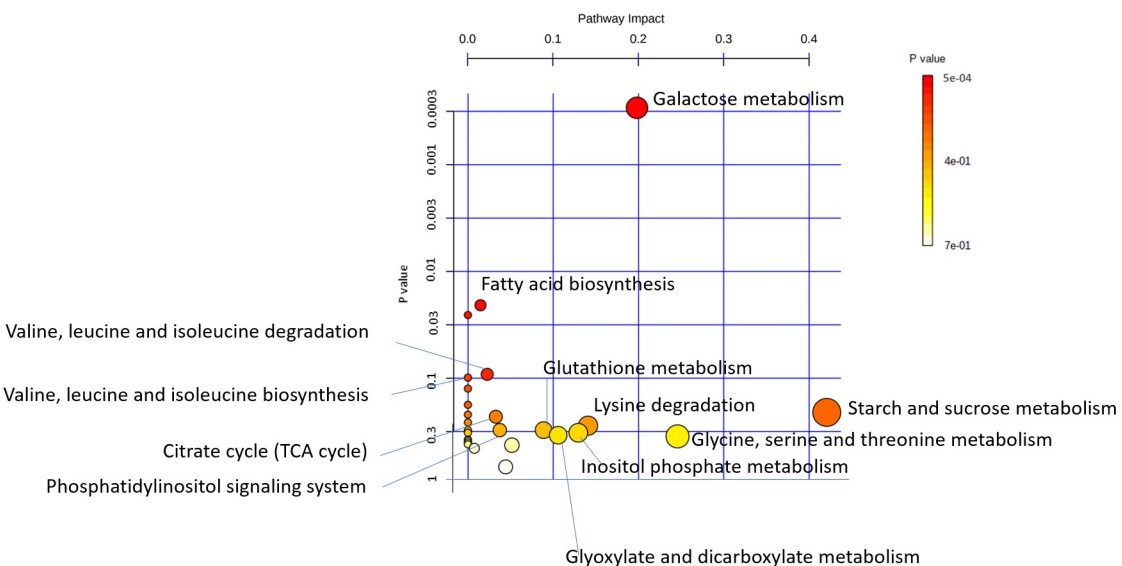

**Fig 7. Pathway analysis of metabolites associated with AVS.** Outputs of urine and plasma metabolomic profiling in AVS patients and control subjects were used to identify changes in biological pathways in the human Kyoto Encyclopedia of Genes and Genomes (KEGG, www.genome.jp/kegg) using the MetaboAnalyst web-tool (www.metaboanalyst.ca). Data are plotted to illustrate the most significantly altered pathways in terms of p-values derived from hypergeometric test on the vertical axis and impact on the horizontal axis.

amino acid glycine, D-Glucose and trans-Aconitic acid were higher in AVS patients than in controls (**Fig 6, Table 3**). Interestingly, evidence of replicated association to AVS in both urine and plasma was observed for a series of 9 metabolites (myristic acid, trans-Aconitic acid, palmitic acid, estrone, 2-Oxovaleric acid, elaidic acid, 2,4-Di-tert-butylphenol, myo-inositol, quinic acid and an unknown metabolite at RT 14.5mins) (**Tables 2 and 3**). However, only 2,4-Di-tert-butylphenol and trans-Aconitic acid displayed concordant pattern of up- or down-regulation in the two biofluids in AVS patients.

## Fatty acid biosynthesis and galactose metabolism are altered in AVS

To elucidate the biological relevance of metabolites associated with AVS, we carried out pathway analysis using data from urine and plasma metabolites associated with AVS. These metabolites are involved in 28 pathways in KEGG. The most significant pathways underlying AVS risk were the metabolism of galactose (α-lactose, galactose, glucose, myo-inositol) (p = 0.0003; FDR p = 0.025), the biosynthesis of fatty acids and, to a lesser extent, the metabolism of branched chain amino acids (valine, leucine, isoleucine) (**Fig 7**). The metabolism of starch and sucrose and the metabolism of glycine, serine and threonine were also detected.

## Discussion

We report results from paralleled untargeted metabolomic profile analyses of urine and plasma in a cohort of patients with AVS and control subjects that identified individual metabolites, metabolite patterns and biological pathways underlying disease risk. Known and unknown metabolites showed biofluid-specific changes between AVS and control individuals or either conserved or discordant regulation patterns in the two biofluids. Our data provide a solid foundation for the definition of metabolites and biological pathways that may be used as potential biomarkers for AVS diagnosis and prevention, as well as targets for therapeutic applications.

Metabolomic profiling is a powerful hypothesis-free strategy that we applied to the identification of a repository of unknown and known urine and plasma metabolites associated with AVS. Several of these associations concern individual metabolites that show evidence of pathophysiological relevance to heart diseases and therefore potentially to AVS. Metabolomic studies have shown that plasma levels of glycine and myo-inositol are increased in patients with heart failure [20]. Elevated plasma myo-inositol was also reported in primary dilated cardiomyopathy [21] and in a preclinical model of myocardial infarction [22]. Elaidic acid is an unsaturated trans-fatty acid, which was found elevated in the serum of patients with coronary artery disease and positively associated with LDL-cholesterol and triglyceride [23, 24].

Biological pathway analysis pointed to fatty acid metabolism as a prominent mechanism associated with AVS in our study. The underlying metabolites that are the most relevant to cardiovascular diseases were the saturated fatty acids dodecanoic, myristic, stearic and palmitic acids and the unsaturated trans fatty acid elaidic acid. Examples of dietary sources of these fatty acids are hydrogenated vegetable oils (elaidic acid), palm oil, meats, cheeses, butter and dairy products (palmitic acid), animal and vegetable fats, coconut and nutmeg oils (myristic acid). Palmitic acid can also be synthesized in the liver through fatty acid biosynthesis or elongation from myristic acid in the mitochondria. Fatty acids are central to the function of the heart since 50 to 70% of its energy is produced by mitochondrial fatty acid β-oxidation [25]. Epidemiological studies and clinical trials have suggested an association between the intake of saturated fatty acids and the risk of coronary heart disease (CHD) [26–28], even though this link remains contested [29]. For instance, the intake of palmitic, stearic and elaidic acids was correlated to the progression of CHD [30] and was associated with a 9–24% increased risk of CHD [31]. Reducing the dietary palmitic and myristic acid decreased the risk for CHD [32], whereas high consumption of myristic acid was correlated with high mortality due to CHD [33], presumably through the effect of myristic, palmitic and stearic acids on increased platelet aggregation [34]. Palmitic, dodecanoic and myristic acids are the major cholesterol-raising saturated fatty acids [35] and diets rich in these metabolites result in high LDL-cholesterol level and low HDL/LDL cholesterol [36]. Plasma levels of myristic acid are negatively associated with HDL-cholesterol in a population characterized with obesity and metabolic syndrome [37]. These fatty acids may therefore contribute to AVS in our study through their role in lowering HDL and increasing LDL-cholesterol levels [38]. The lack of association between AVS and fatty acids despite non-significant differences in lipoprotein levels between cases and controls may be explained by the treatment of many patients with lipid lowering medications (statins).

Paralleled metabolomic analysis of plasma and urine samples provides information about biofluid-specific changes in AVS. It also identifies conserved associations of metabolites to AVS in the two biofluids, which may suggest dysregulation in relevant biological pathways and allows prediction of the level of plasma metabolites based on their urine concentration in patients. Intriguingly, over 75% of GC-MS features and over 80% of known metabolites associated with AVS were up-regulated in urine and downregulated in plasma. In addition, even though the level of half of these known metabolites was different between patients and controls in both plasma and urine, they were almost all upregulated in urine and downregulated in plasma in the AVS group, suggesting a stimulation of their elimination in urine of patients. Only 2,4-Di-tert-butylphenol and trans-Aconitic acid (TAA) displayed consistent trend of regulation in the two biofluids.

2,4-Di-tert-butylphenol is a lipophilic phenol present in the environment and a product of bacterial metabolism [39]. It shows antioxidant properties against LDL-oxidation [40], thus potentially preventing atherosclerosis, and anti-inflammatory properties by decreasing the expression of TNF-α, interleukins *IL-6* and *IL-1b* in a mouse macrophage cell line [41]. TAA is

an unsaturated tricarboxylic acid and an isomer of the tricarboxylic acid cycle intermediate cis-Aconitic acid. TAA is mainly obtained from the sugar cane molasses [42] and is also metabolized by bacteria [43]. Metabolomic studies in humans indicated that the isomer cis-Aconitic acid is downregulated in the plasma of patients with CHD [44] and that aconitic acid is a marker of myocardial injury [45]. These data support a role of 2,4-Di-tert-butylphenol and TAA in AVS, which requires experimental validation.

## Conclusions

Our findings provide initial evidence of candidate metabolite biomarkers of AVS and raise novel hypotheses regarding their contribution to the disease and the relevant pathophysiological mechanisms involved. Many metabolites associated with AVS in our study are involved in biological pathways that do not have obvious relevance to heart diseases, and therefore open new research avenues to test their implication in AVS etiopathogenesis. In addition, several urine and plasma metabolomic features associated with AVS remain unknown and require chemical attribution for unambiguous identification of the underlying metabolites. Further investigations are warranted to replicate association between metabolites and AVS in larger population studies. In addition, future work is required to determine whether the candidate metabolites identified affect aortic valve either directly or indirectly through factors known to contribute to AVS risk, including for example cholesterol metabolism and Lpa. We have verified that metabolites associated with AVS do not show evidence of significant association with plasma LDL and HDL (**S4 Fig**). Along the same line, changes in these metabolites may be reactive to AVS pathology and drug treatments. Assessment of causal relationships between the candidate metabolites that we have identified and AVS can be tested through extended genetic association analyses and application of Mendelian randomization methods in large genetic studies.

## Supporting information

**S1 Table. Clinical and biochemical features in males and females selected for presence or absence of aortic valve stenosis.** Data are means ± SEM.
(DOCX)

**S2 Table. Metabolites identified using the NIST08 library (https://chemdata.nist.gov/) after analysis of gas chromatography mass spectrometry (GC-MS) of urine and plasma samples of patients with aortic valve stenosis and controls.** RT, Retention Time; HMDB, Human Metabolome Database; KEGG, Kyoto Encyclopedia of Genes and Genomes.
(XLSX)

**S3 Table. Urinary and plasma metabolites contributing to the separation between the AVS patients and healthy controls.** Data were derived by GC-MS analysis of urine and plasma samples from 46 patients and 46 controls. Data were normalized to the internal standard 2-isopropylmalic acid. Variable importance in the projection (VIP) was obtained from PLS-DA with a threshold of 1.0; p-values are calculated from a volcano plot; q-values are the adjusted p-value with Benjamini-Hochberg method. Area Under the Curve (AUC) was calculated using the online tool MetaboAnalyst to determine biomarker utility. Regulation gives information on up- or down-regulation of the features in AVS patients. RT, Retention time; FDR, False Discovery Rate.
(XLSX)

**S4 Table. Urinary metabolites contributing to the separation between the AVS patients and controls.** Data were derived by GC-MS analysis of urine samples from 46 patients and 46

controls. Data were normalized to creatinine and logTranformed. Variable importance in the projection (VIP) was obtained from PLS-DA with a threshold of 1.0; q-values are the adjusted p-value with Benjamini-Hochberg method. Regulation gives information on up- or down-regulation of the features in AVS patients. RT, Retention time; FDR, False Discovery Rate.
(XLSX)

**S5 Table. Urine metabolic fingerprint of AVS patients and healthy controls.** Intensity values derived by GC-MS analysis of urine samples in AVS cases and controls were normalized to the internal standard 2-isopropylmalic acid.
(XLSX)

**S6 Table. Plasma metabolic fingerprint of AVS patients and healthy controls.** Intensity values derived by GC-MS analysis of plasma samples in AVS cases and controls were normalized to the internal standard 2-isopropylmalic acid.
(XLSX)

**S1 Fig. Validation of the OPLS-DA model of biofluid metabolomic data from patients with aortic valve stenosis (AVS) and control individuals.** Data were derived from GC-MS spectra of urine (a, b) and plasma (c, d) samples from AVS patients (n = 46) and control individuals (n = 46). Model validation was performed using permutation test with 1000 iterations on the OPLS-DA model. Empirical p-values Q2: $p<0.001$ and R2Y: $p<0.001$.
(TIF)

**S2 Fig. ROC analysis of candidate urine metabolites separating AVS patients and control individuals.** Each of the 16 candidate metabolites (VIP>1, nominal $p<0.05$, $q<0.05$) has a ROC curve where the sensitivity is on the y-axis and the specificity is on the x-axis. The AUROC is shown in blue and the AUC values with their 95% confidence intervals are presented in the curves.
(TIF)

**S3 Fig. ROC analysis of candidate plasma metabolites separating AVS patients and control individuals.** Each of the 14 candidate metabolites (VIP>1, nominal $p<0.05$, $q<0.05$) has a ROC curve where the sensitivity is on the y-axis and the specificity is on the x-axis. The AUROC is shown in blue and the AUC values with their 95% confidence intervals are presented in the curves.
(TIF)

**S4 Fig. Association analysis of plasma GC-MS spectral data with HDL and LDL in patients and controls.** Data were derived by GC-MS analysis of plasma samples from 46 AVS patients and 46 control individuals. Generalized linear models were used to determine significant associations between metabolomic peaks and HDL (a) and LDL (b) and correcting for multiple testing. Signal intensities normalized to the internal standard are plotted against the Q-values.
(TIF)

## Author Contributions

**Conceptualization:** Rony S. Khnayzer, Dominique Gauguier, Pierre A. Zalloua.

**Data curation:** Georges Khazen, Dominique Gauguier.

**Formal analysis:** Cynthia Al Hageh, Ryan Rahy, Georges Khazen, Francois Brial.

**Funding acquisition:** Rony S. Khnayzer, Dominique Gauguier, Pierre A. Zalloua.

**Investigation:** Cynthia Al Hageh, Francois Brial.

**Methodology:** Dominique Gauguier.

**Resources:** Rony S. Khnayzer, Pierre A. Zalloua.

**Supervision:** Rony S. Khnayzer, Dominique Gauguier, Pierre A. Zalloua.

**Writing – original draft:** Dominique Gauguier, Pierre A. Zalloua.

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
