## [Decision Letter · Decision Letter 0]

9 Sep 2020

PONE-D-20-24273

Plasma and urine metabolomic analyses in aortic valve stenosis reveal shared and biofluid-specific metabolic regulation

PLOS ONE

Dear Dr. Gauguier,

Thank you for submitting your manuscript to PLOS ONE. After careful consideration, we feel that it has merit but does not fully meet PLOS ONE’s publication criteria as it currently stands. Therefore, we invite you to submit a revised version of the manuscript that addresses the points raised during the review process.

The paper is of quite some interest, but, as evident from the reviewer comments, multiple detail require attention, on several occasions more clarity/a (better) explanation appears necessary. However, due to the excellent comments, this should not present a major obstacle.

We look forward to receiving your revised manuscript.

Kind regards,

Harald Mischak

Academic Editor

PLOS ONE

Journal Requirements:

Reviewers' comments:

Reviewer's Responses to Questions

**Comments to the Author**

1. Is the manuscript technically sound, and do the data support the conclusions?

Reviewer #1: Yes

Reviewer #2: Yes

Reviewer #3: Partly

2. Has the statistical analysis been performed appropriately and rigorously? 

Reviewer #1: Yes

Reviewer #2: Yes

Reviewer #3: Yes

3. Have the authors made all data underlying the findings in their manuscript fully available?

Reviewer #1: No

Reviewer #2: Yes

Reviewer #3: Yes

4. Is the manuscript presented in an intelligible fashion and written in standard English?

Reviewer #1: Yes

Reviewer #2: Yes

Reviewer #3: Yes

5. Review Comments to the Author

Reviewer #1: Overall, the manuscript is well written presenting results and conclusions supported by the data. Few minor comments:

- The text in figure 1 is not readable. It would be helpful if the authors could use a bigger font in all parts. Specifically, it would be better if parts d and h could be moved to another figure(s), as the bar graphs would be more clear in this case and the rest of the plots would be bigger.

- Figure 4 could be separated in two or more figures so that the bars and the text are more clear.

- All data should be provided as part of the manuscript or its supporting information, or deposited to a public repository.

Reviewer #2: The article from Al Hageh et al. describes the significant changes in plasma and urine metabolomics in patients with aortic valve stenosis, compared to matched controls. The study is methodologically sound, statistically appropriate and results are cautiously interpreted. However, the article appears to fall in between two study types that are biomarker discovery and physiopathology. By focusing on one or the other, the study would have more impact. The results on 2,4-di-tert-butylphenol and Trans aconitic acid are of the greatest interest (fig 4a).

A major missing information in the study is a technical definition of AVS diagnosis (in general and in the study) and a summary of AVS-related echography results from cases and controls. Could a severity score be obtained and correlated to metabolomics results ?

The result section shows a promising discriminating urinary pattern in PCA and OPLS-DA. This could define a multidimensional biomarker of AVS, to be tested in a validation cohort or compared to an AVS severity score. Correlations between individual candidate biomarker levels and AVS severity scores could also be interesting.

The discussion on the biological interpretation of observed changes would gain from being more in depth. What is the link between dietary and circulating fatty acids ? Could a pathway analysis be relevant to the study?

The authors report changes in urine levels of many substances between AVS patients and controls, and conclude to changes in renal elimination. Did the authors control the presence of CKD (eGFR to add to table S1) and its influence by normalizing to urine creatinine level?

One can get lost in the large number of data, tables, figures and supplements that are provided (which should be simplified). For instance, all AUCs (p11-12) and estimates (from logistic regression? For what unit change?) could be placed in a table, possibly containing results from different successive steps. This would help readers go through the result section and provide a general view of results.

Other comments:

- How were the blood and urine samples obtained (At what time of the day ? In fasting conditions ?), prepared and stored?

- For molecules with discriminant ability, the cutoff values could be reported

- The wording “differential (metabolic) regulations” is largely used while misleading, particularly in the title. Using simple terms such as “different levels” would be better.

- The figures quality is low and prevents any interpretation of their content (blurry, unreadable). The figure legends should explain all the items depicted in figures. In addition:

• Figure 1a/e: What are the different colours?

• Figure 1f: The overlapping of clusters in PCA and OPLS-DA is in contrast with the statement of “clearly independent clustering” (line 181, p10)

• Figure 2-3: Please explain why a given metabolite can have several features. Shouldn’t metabolites with various TMS adducts be summed up ?

• Figure 4. Please reword/simplify the figure title to match its content. What are the units of metabolite levels?

- Table S1: The percentages in different columns must be related to the sample size of that column. This table (or parts of it) deserve to be in the main manuscript.

- Line 272: genomic strategy?

Reviewer #3: This interesting article by Gauguier et al describe both plasma and urinary metabolite profiles associated with Aortic Valve Stenosis. Contrasting trends in metabolic regulation were noted between the two bio-fluids. I do have a few comments:

1. In the abstract reference is made to “pre-processed data” what is meant with this?

2. I think the abstract can benefit from adding specifics regarding the metabolites that stood out in this study. In the current state you have to read all the way to the results sections of the paper to get an idea about the metabolic pathways affected by AVS.

3. Spacing before in text references should be corrected.

4. In the introduction line 43-44: I think the two sentences can be combined. The second statement needs a reference.

5. Try to avoid repetition e.g. line 51-52 “Metabolomics has been extensively used for biomarker discovery, drug response ascertainment and disease pathway identification” and line 54-55 “ Metabolomics provides a platform for biomedical discovery as well as clinical and pharmaceutical applications”

6. The end of the introduction reads as an abstract in that it is ending with a vague idea of what was found in the study (line 61-68). I think it would be better to just end the introduction with the aim of the study.

7. In the methods section a lot of basic results is given. I would suggest that his is moved to the results sections.

8. Normally a sentence would not start with a number (e.g. 63 subjects), but rather with the number written in words.

9. It is stated in the text that there is no differences between AVS patients and controls for biochemical variables (line 82-84 page 5) but it think Table S1 should also indicate p-values between groups to indicate this.

10. Some abbreviations should be explained upon first mention, and normally a sentence should not start with an abbreviation.

11. In the results section (page 9 line 145) it is stated that 190 and 130 peaks have been confidently detected in urine and plasma. Perhaps just add a measure of confidence there.

12. The levels of urinary metabolites is determined by the concentration of urine. Did you adjust these levels of creatinine?

13. Also the opposing findings in plasma vs urine may have something to do with renal function. Do you have any information on renal function of the AVS vs controls?

14. The results section on page 13 (line 235 – 248) is difficult to follow. Also reference is made to Table 1a and Table 1b, but Table 1 as included in the article does not have a “a” and “b” part.

15. Apart from just comparing metabolic profiles (urine and plasma) between AVS and controls perhaps you can also look into correlations of significantly different metabolites with echo parameters?

16. Figure 4 is nice to indicate the differences between the bio-samples and AVS vs controls. But perhaps you can also consider a Venn diagram to indicate differences and similarities?

17. I think the discussion can benefit greatly from a figure of the metabolic pathways identified.

18. The involvement of fatty acids is interesting but also expected as fatty acids are the main source of energy in the heart, perhaps you can just add some reference to that in the discussion.

19. In this study the lipid profile (Table S1) did not differ between the AVS and controls but yet the fatty acids are quite prominent. Also when considering that you mentioned in the Conclusions that there was no associations between LDL and HDL (results not shown) Perhaps you can comment on that?

6. PLOS authors have the option to publish the peer review history of their article (what does this mean?). If published, this will include your full peer review and any attached files.

Reviewer #1: No

Reviewer #2: No

Reviewer #3: **Yes: **Catharina M.C. Mels

---

## [Author Response · Author response to Decision Letter 0]

1 Oct 2020

Detailed responses are in the file "PONE-D-20-24273_Response to Reviewers" and below:

Reviewer #1: Overall, the manuscript is well written presenting results and conclusions supported by the data. Few minor comments:

We thank the Reviewer for her/his review of our manuscript.

- The text in figure 1 is not readable. It would be helpful if the authors could use a bigger font in all parts. Specifically, it would be better if parts d and h could be moved to another figure(s), as the bar graphs would be more clear in this case and the rest of the plots would be bigger.

Response: We thank the Reviewer for this suggestion. We have split Figure 1 in two figures (new Fig1= old Fig 1a,b,c,e,f,g; new Fig2= old Fig1d+h) and revised the text accordingly.

- Figure 4 could be separated in two or more figures so that the bars and the text are more clear.

Response: As suggested by the Reviewer, the text on the figure has been enhanced and Figure 4 has been split in two figures (Figs 5 and 6 in the revised version of the manuscript). Reference to the figures has been amended in the text accordingly.

- All data should be provided as part of the manuscript or its supporting information, or deposited to a public repository.

Response: As requested by the Reviewer, the raw datafiles of the metabolomic profiling experiments in urine and plasma samples (GC/MS data normalised to the internal standard) have been included in Supplementary tables 4 (urine) and 5 (plasma).

Reviewer #2: The article from Al Hageh et al. describes the significant changes in plasma and urine metabolomics in patients with aortic valve stenosis, compared to matched controls. The study is methodologically sound, statistically appropriate and results are cautiously interpreted. However, the article appears to fall in between two study types that are biomarker discovery and physiopathology. By focusing on one or the other, the study would have more impact. The results on 2,4-di-tert-butylphenol and Trans aconitic acid are of the greatest interest (fig 4a).

We thank the Reviewer for her/his positive evaluation of our manuscript and her/his useful suggestions.

A major missing information in the study is a technical definition of AVS diagnosis (in general and in the study) and a summary of AVS-related echography results from cases and controls. Could a severity score be obtained and correlated to metabolomics results ?

Response: The Reviewer raises an important point that we now address in the Methods section of the paper. Aortic valve stenosis was diagnosed initially on physical examination by finding a systolic murmur in the aortic valve area. It was subsequently confirmed by Echocardiography. In our study, AVS was considered present if reported by the clinician in the Medical Record. However, the degree and severity of AVS were not recorded and could not be included in our database. This is now clarified in the Methods section, page 5, lines 76-78 of the revised version of the manuscript.

The result section shows a promising discriminating urinary pattern in PCA and OPLS-DA. This could define a multidimensional biomarker of AVS, to be tested in a validation cohort or compared to an AVS severity score. Correlations between individual candidate biomarker levels and AVS severity scores could also be interesting.

Response: As above mentioned, only presence or absence of AVS was reported in subjects showing systolic murmur in the aortic valve area. Unfortunately, the degree and severity of AVS were not recorded at the time of clinical examination and were therefore not available in our database.

The discussion on the biological interpretation of observed changes would gain from being more in depth. What is the link between dietary and circulating fatty acids ? Could a pathway analysis be relevant to the study?

Response: As suggested by the Reviewer, we have addressed the point about dietary fatty acids in the discussion (Page 15-16, lines 299-303).

The second point made by Reviewer 2 about the need to carry out pathway analysis was also raised by Reviewer 3. To comply with these suggestions, we carried out this analysis using MetaboAnalyst and the KEGG database and identified predominantly changes in fatty acid biosynthesis and galactose metabolism are altered in AVS. Data are now reported at the end of the result section of the revised version of the manuscript (Page 14, lines 270-277). Methods are described on Page 8 (lines 143-150). An additional figure (Figure 7) illustrates results from pathway analysis.

The occurrence of fatty acid metabolism in results from pathway analysis is relevant to the point made by the Reviewer about dietary and circulating fatty acids. As suggested by Reviewer 2, we developed the discussion with biological interpretation of the data, with a specific focus on fatty acids (page 15-16, lines 296-304).

The authors report changes in urine levels of many substances between AVS patients and controls, and conclude to changes in renal elimination. Did the authors control the presence of CKD (eGFR to add to table S1) and its influence by normalizing to urine creatinine level?

Response: The Reviewer is correct that adjustment for several confounding factors is required to analyse urine metabolome data. We could not control for the presence of CKD since eGFR values were missing for many patients, hence if included would have significantly reduced the power of the analyses. Urine data were normalised to the internal standard 2-isopropylmalic acid, which prevented subsequent normalisation to creatinine.

One can get lost in the large number of data, tables, figures and supplements that are provided (which should be simplified). For instance, all AUCs (p11-12) and estimates (from logistic regression? For what unit change?) could be placed in a table, possibly containing results from different successive steps. This would help readers go through the result section and provide a general view of results.

Response: As suggested by the Reviewer, we have simplified the report of data in the result section and that of technical details in the supplementary material. We have merged supplementary Tables 2 and 3 in a single table (now supplementary Table 2). Along the same line, supplementary Tables 4 and 5 have been merged in a single table (now supplementary Table 3). In the latter, the metabolites have been ranked according to their VIP instead of their retention time, in order to underline those contributing the most significantly to the separation between cases and controls. The text on page 11 has been simplified by moving the AUC and 95% CI for associated metabolites in supplementary table 3. The text on page 12 has also been simplified for metabolites associated with AVS after correction for confounders: we removed from the text values of the regression coefficients and confidence intervals which are already given in Tables 2 (urine) and 3 (plasma) (Table 1 has been split in two tables in response to a comment from Reviewer 3.

- How were the blood and urine samples obtained (At what time of the day ? In fasting conditions ?), prepared and stored?

Response: As requested by the Reviewer, information about sample collection and storage has been added in the methods section of the revised version of the manuscript, page 5, lines 80-83. About 30 ml of urine and 20 ml of arterial blood was collected after 12 hours fasting. Blood collected in EDTA tubes was spun for plasma separation. Plasma and urine aliquots were stored at -80oC until metabolomic analysis.

- For molecules with discriminant ability, the cutoff values could be reported.

Response: As mentioned in the methods section page 7, lines 134-136, metabolites with discriminant potential were ranked as excellent (AUC=0.9-1.0), good (AUC=0.8-0.9), fair (AUC=0.7-0.8), poor (AUC=0.6-0.7) or failed (AUC=0.5-0.6). For clarity, this information has been added in the text on page 12, lines 227-240.

- The wording “differential (metabolic) regulations” is largely used while misleading, particularly in the title. Using simple terms such as “different levels” would be better.

Response: As requested by the Reviewer, the suggested changes have been made throughout the text where appropriate, as well as in the title of the paper which now reads: “Plasma and urine metabolomic analyses in aortic valve stenosis reveal shared and biofluid-specific changes in metabolite levels”. 

- The figures quality is low and prevents any interpretation of their content (blurry, unreadable). The figure legends should explain all the items depicted in figures. In addition:

Response: As suggested by the Reviewer, the text on figures has been enhanced and their resolution has been increased.

• Figure 1a/e: What are the different colours?

Response: We apologise to the Reviewer if the color code on fig1a/e (now fig1a/d) was unclear. We did mention in the figure legend that pink dots in the upper part of the volcano plots indicated metabolomic features that significantly separated cases and controls (nominal p<0.05). We have rephrased the text of the legend to clarify this point.

• Figure 1f: The overlapping of clusters in PCA and OPLS-DA is in contrast with the statement of “clearly independent clustering” (line 181, p10)

Response: We thank the Reviewer for pointing this. We have amended the text to underline that PCA-based clustering of plasma data was inferior to that of urine data.

• Figure 2-3: Please explain why a given metabolite can have several features. Shouldn’t metabolites with various TMS adducts be summed up?

Response: In this study we used the XCMS tool for data pre-processing to correct deviations in the retention time from one sample to the next. XCMS uses two pieces of information to do the alignment: RT and the highest point in MS/MS spectrum of each RT. Thus, the feature detected for example at RT=7.6 has a MS/MS spectrum with three major peaks at m/z 72, 147 and 275. Thus, if in one sample the highest peak in the MS/MS spectrum is at m/z 72 and in another sample the highest peak is at m/z 275, in this case XCMS considers them as two separate peaks and it doesn’t align them as one peak. For that reason we can find many peaks for the same metabolite.

• Figure 4. Please reword/simplify the figure title to match its content. What are the units of metabolite levels?

Response: We thank the Reviewer for pointing to this detail. The units are intensities of MS signals for each metabolite normalised by that of the internal standard 2-isopropylmalic acid. As requested, the title of figure 4 (now Figures 5 and 6) has been simplified. We have clarified in the legend to the figure that the unit are log transformed intensities normalised to the internal standard.

- Table S1: The percentages in different columns must be related to the sample size of that column. This table (or parts of it) deserve to be in the main manuscript.

Response: We thank the Reviewer for spotting errors in the calculation of percentages for cases and controls, which have been rectified in Table S1 and in the new Table 1. As suggested by the Reviewer, part of Table S1 has been moved to the main manuscript. It is now Table 1 in the revised version of the manuscript. This table reports data in cases and controls, whereas data in males and females remain in Table S1.

- Line 272: genomic strategy?

Response: “genomic” has been withdrawn.

Reviewer #3: This interesting article by Gauguier et al describe both plasma and urinary metabolite profiles associated with Aortic Valve Stenosis. Contrasting trends in metabolic regulation were noted between the two bio-fluids.

The authors thank the Reviewer for her thorough evaluation of our manuscript and her useful comments and suggestions.

I do have a few comments:

1. In the abstract reference is made to “pre-processed data” what is meant with this? 

Response: The pre-processing stage involved merging all the raw MS data in one peak table, which requires detection and alignment of the same peaks for all the sample prior to statistical analysis. Technically, the treatment of the raw data in order to get one table (dataMatrix) that contain all the samples with the aligned peaks is called pre-processing. After this step comes the processing where the chemometric analysis is applied such as PCA, PLS-DA, This point has been clarified in the abstract of the revised version of the manuscript.

2. I think the abstract can benefit from adding specifics regarding the metabolites that stood out in this study. In the current state you have to read all the way to the results sections of the paper to get an idea about the metabolic pathways affected by AVS.

Response: We agree with the Reviewer that the abstract lacks details about metabolites associated with AVS. As suggested, we have added towards the end of the abstract details of metabolites associated with AVS that show shared or discordant pattern of regulation in plasma and urine.

3. Spacing before in text references should be corrected. 

Response: This has been corrected throughout the text. 

4. In the introduction line 43-44: I think the two sentences can be combined. The second statement needs a reference.

Response: We thank the Reviewer for this suggestion. We have changed the text accordingly (Page 3, Lines 47-48).

5. Try to avoid repetition e.g. line 51-52 “Metabolomics has been extensively used for biomarker discovery, drug response ascertainment and disease pathway identification” and line 54-55 “Metabolomics provides a platform for biomedical discovery as well as clinical and pharmaceutical applications”

Response: We agree with the Reviewer’s comment. As suggested we have merged the two sentences in the revised version of the manuscript (Page 3, Lines 55-57).

6. The end of the introduction reads as an abstract in that it is ending with a vague idea of what was found in the study (line 61-68). I think it would be better to just end the introduction with the aim of the study.

Response: As suggested we have changed the end of the introduction by providing the general objectives of the study rather than results.

7. In the methods section a lot of basic results is given. I would suggest that his is moved to the results sections.

Response: This point was also pointed by Reviewer 2. To follow the recommendation of the two Reviewers, we have moved the description of the biochemical and clinical data in patients and control subjects from the methods to results section. Also part of Table S1 (data in cases and controls) has been moved to the main text as a new table (Table 1). Data in males and females remain in Table S1.

8. Normally a sentence would not start with a number (e.g. 63 subjects), but rather with the number written in words.

Response: This has been rectified (Page 9, line 157).

9. It is stated in the text that there is no differences between AVS patients and controls for biochemical variables (line 82-84 page 5) but it think Table S1 should also indicate p-values between groups to indicate this.

Response: As requested, a column with P-values has been added to TableS1 (Table 1 in the revised version of the manuscript).

10. Some abbreviations should be explained upon first mention, and normally a sentence should not start with an abbreviation.

Response: we have been through the manuscript and made corrections accordingly.

11. In the results section (page 9 line 145) it is stated that 190 and 130 peaks have been confidently detected in urine and plasma. Perhaps just add a measure of confidence there.

Response: The Reviewer is correct that threshold of detection of peaks should be mentioned in the manuscript. Following verifications with the equipment supplier and the platform, we can confirm that a signal to noise ratio of 6 was used to as detection threshold. This is now reported in the methods section (Page 9, line 165). 

12. The levels of urinary metabolites is determined by the concentration of urine. Did you adjust these levels of creatinine?

Response: All urine and plasma MS data were systematically normalised to the internal standard 2-isopropylmalic acid. To ensure consistency in statistical analysis of plasma and urine data, we did not adjust the urine data for creatinine levels. Adjusting urine data for creatinine would have resulted in a normalisation for two variables. In our hands, this has resulted to very low values and was problematic for statistical analysis.

13. Also the opposing findings in plasma vs urine may have something to do with renal function. Do you have any information on renal function of the AVS vs controls?

Response: This important point was also raised by Reviewer 2. eGFR values were missing for many patients in our dataset, which prevented their use in the analysis since it would have significantly reduced the power of the analyses, and hampered the reliability of results.

14. The results section on page 13 (line 235 – 248) is difficult to follow. Also reference is made to Table 1a and Table 1b, but Table 1 as included in the article does not have a “a” and “b” part.

Response: To clarify statistical results in urine and plasma initially reported in Tables 1a and 1b, the table has been split in two tables (table 1 for urine data and table 2 for plasma data).

15. Apart from just comparing metabolic profiles (urine and plasma) between AVS and controls perhaps you can also look into correlations of significantly different metabolites with echo parameters?

Response: The Reviewer raises and important point. Unfortunately, in our database, AVS was considered present if reported by the clinician in the Medical Record. We had no access to the echo parameters of the patients.

16. Figure 4 is nice to indicate the differences between the bio-samples and AVS vs controls. But perhaps you can also consider a Venn diagram to indicate differences and similarities?

Response: We agree with the Reviewer that a Venn diagram would be a useful illustration of our results. On the other hand, plotting the intensity values provides the reader with better estimates of means and inter-individual variability, which is the reason we chose to represent the results as box plots.

17. I think the discussion can benefit greatly from a figure of the metabolic pathways identified.

Response: We agree with the Reviewer that pathway analysis would be an interesting addition in the manuscript. We carried out this analysis using MetaboAnalyst and the KEGG database and identified predominantly changes in fatty acid biosynthesis and galactose metabolism are altered in AVS. This is now reported at the end of the result section (Page 14, lines 270-277). Methods are described in the relevant section on Page 8 (lines 143-150). An additional figure (Figure 7) illustrates results from pathway analysis.

18. The involvement of fatty acids is interesting but also expected as fatty acids are the main source of energy in the heart, perhaps you can just add some reference to that in the discussion.

Response: The point raised by Reviewer 3 about the need to improve the discussion with additional information on fatty acids was also raised by Reviewer 2. We agree that this is an important aspect in the article, which is supported by results from pathway analysis. As requested we have developed the discussion along these lines (Page 16, lines 303-304). A reference (24- Lopaschuk et al. Myocardial Fatty Acid Metabolism in Health and Disease. Physiological Reviews 90, 207–258,2010) has been added.

19. In this study the lipid profile (Table S1) did not differ between the AVS and controls but yet the fatty acids are quite prominent. Also when considering that you mentioned in the Conclusions that there was no associations between LDL and HDL (results not shown) Perhaps you can comment on that?

Response: The Reviewer is correct that association between AVS and fatty acids are quite prominent in our study despite lack of associations with lipoproteins. This lack of association maybe partly due to the fact that most of the patients were under lipid lowering medications (Statins), hence any association would have been very difficult to ascertain, especially that adjusting for medication use could not be performed due to loss of power. As suggested by the Reviewer, we have revised the discussion accordingly (Page 16, lines 317-319). Also, as requested by the journal we have added a supplementary figure (S4 Fig) to show the lack of association between metabolite features and LDL and HDL.

---

## [Decision Letter · Decision Letter 1]

19 Oct 2020

PONE-D-20-24273R1

Plasma and urine metabolomic analyses in aortic valve stenosis reveal shared and biofluid-specific changes in metabolite levels

PLOS ONE

Dear Dr. Gauguier,

Thank you for submitting your manuscript to PLOS ONE. After careful consideration, we feel that it has merit but does not fully meet PLOS ONE’s publication criteria as it currently stands. Therefore, we invite you to submit a revised version of the manuscript that addresses the points raised during the review process.

We look forward to receiving your revised manuscript.

Kind regards,

Harald Mischak

Academic Editor

PLOS ONE

Additional Editor Comments (if provided):

Please perform the minor changes requested by reviewer 2 and resubmit, so the paper can be accepted.

Reviewers' comments:

Reviewer's Responses to Questions

**Comments to the Author**

1. If the authors have adequately addressed your comments raised in a previous round of review and you feel that this manuscript is now acceptable for publication, you may indicate that here to bypass the “Comments to the Author” section, enter your conflict of interest statement in the “Confidential to Editor” section, and submit your "Accept" recommendation.

Reviewer #1: All comments have been addressed

Reviewer #2: All comments have been addressed

Reviewer #3: All comments have been addressed

2. Is the manuscript technically sound, and do the data support the conclusions?

Reviewer #1: Yes

Reviewer #2: Yes

Reviewer #3: Yes

3. Has the statistical analysis been performed appropriately and rigorously? 

Reviewer #1: Yes

Reviewer #2: Yes

Reviewer #3: Yes

4. Have the authors made all data underlying the findings in their manuscript fully available?

Reviewer #1: Yes

Reviewer #2: Yes

Reviewer #3: Yes

5. Is the manuscript presented in an intelligible fashion and written in standard English?

Reviewer #1: Yes

Reviewer #2: Yes

Reviewer #3: Yes

6. Review Comments to the Author

Reviewer #1: (No Response)

Reviewer #2: The authors have answered all comments, but one: the addition of a correction for urine dilution is feasible and would have been highly relevant to the study.

Minor comments:

*Lines 264-268: the two sentences appear contradictory. Please edit the first one to clarify that the series of metabolites are consistent within (not across) fluids.

*Figures 5 and 6:

-What is the y-axis legend ? what units? what numbers (impossible to read)?

-Display the controls on the left and cases on the right to help readers

*Figure 7:

-identify starch metabolism

-display backtransformed P-values on the y-axis (and color legend).

Reviewer #3: Thank you for taking the time to make the requested amendments. I have nothing further to add.

7. PLOS authors have the option to publish the peer review history of their article (what does this mean?). If published, this will include your full peer review and any attached files.

Reviewer #1: No

Reviewer #2: No

Reviewer #3: No

---

## [Author Response · Author response to Decision Letter 1]

23 Oct 2020

Comments to the Author

1. If the authors have adequately addressed your comments raised in a previous round of review and you feel that this manuscript is now acceptable for publication, you may indicate that here to bypass the “Comments to the Author” section, enter your conflict of interest statement in the “Confidential to Editor” section, and submit your "Accept" recommendation.

Reviewer #1: All comments have been addressed

Reviewer #2: All comments have been addressed

Reviewer #3: All comments have been addressed

2. Is the manuscript technically sound, and do the data support the conclusions?

Reviewer #1: Yes

Reviewer #2: Yes

Reviewer #3: Yes

3. Has the statistical analysis been performed appropriately and rigorously? 

Reviewer #1: Yes

Reviewer #2: Yes

Reviewer #3: Yes

4. Have the authors made all data underlying the findings in their manuscript fully available?

Reviewer #1: Yes

Reviewer #2: Yes

Reviewer #3: Yes

5. Is the manuscript presented in an intelligible fashion and written in standard English?

Reviewer #1: Yes

Reviewer #2: Yes

Reviewer #3: Yes

6. Review Comments to the Author

Reviewer #1: (No Response)

Reviewer #2: The authors have answered all comments, but one: the addition of a correction for urine dilution is feasible and would have been highly relevant to the study.

Response: We agree with the Reviewer that urine dilution is an important criterion when analysing urine metabolomics. As a rule, we also have to deal with correction for technical deviation between samples and runs using the internal standard in order to consistently normalise the urine and plasma datasets with exactly the same method.

As requested by the Reviewer, we have recalculated the statistics using urine data normalised to creatinine instead of the internal standard. We have mentioned this in the methods section.

The results from association studies between urine metabolites and AVS are very concordant in both cases of data normalisation. We now report in a new table in the revised version of the manuscript (Supplementary Table 4) results from the association between AVS and urine metabolites normalised to creatinine before adjustment for age, sex, body mass index, hyperlipidemia and diabetes.

Also, we have changed the format of Table 2 to show in parallel association results, following adjustment for age, sex, body mass index, hyperlipidemia and diabetes, based on urine metabolomic data independently normalised to the internal standard and to creatinine. This provides the reader with elements to directly compare association data using these two methods, and to be able to visualise consistency in the outputs. Essentially, almost all associations identified following normalisation to the internal standard are replicated when the data are normalised to creatinine, suggesting higher stringency upon data normalisation to the internal standard.

Details of these new findings are given in the results section, pages 11 to 14. 

A reference (Fiehn O. Metabolomics by Gas Chromatography-Mass Spectrometry: Combined Targeted and Untargeted Profiling. Curr Protoc Mol Biol. 2016;114:30.4.1-.4.2) was added. 

Minor comments:

*Lines 264-268: the two sentences appear contradictory. Please edit the first one to clarify that the series of metabolites are consistent within (not across) fluids.

Response: The reviewer is correct. The sentences have been changed (lines 288-293 of the revised manuscript).

*Figures 5 and 6:

-What is the y-axis legend ? what units? what numbers (impossible to read)?

-Display the controls on the left and cases on the right to help readers

Response: Figures 5 and 6 have been redrawn as requested by the Reviewer. Font has been increased and data from controls and cases are shown on the left and right, respectively, for each panel. Details of the unit of the Y-axis are given in the legends to the figures.

*Figure 7:

-identify starch metabolism

Response: As requested, starch metabolism, which was omitted on the original version of Figure 7, was added in the revised version of this Figure. The text on page 15 was amended. We apologise for this mistake.

-display backtransformed P-values on the y-axis (and color legend).

Response: As requested, the revised version of Figure 7 displays backtransformed P-values on the y-axis.

Reviewer #3: Thank you for taking the time to make the requested amendments. I have nothing further to add.

7. PLOS authors have the option to publish the peer review history of their article (what does this mean?). Do you want your identity to be public for this peer review? For information about this choice, including consent withdrawal, please see our Privacy Policy.

Reviewer #1: No

Reviewer #2: No

Reviewer #3: No

---

## [Editor Report · Decision Letter 2]

26 Oct 2020

Plasma and urine metabolomic analyses in aortic valve stenosis reveal shared and biofluid-specific changes in metabolite levels

PONE-D-20-24273R2

Dear Dr. Gauguier,

We’re pleased to inform you that your manuscript has been judged scientifically suitable for publication and will be formally accepted for publication once it meets all outstanding technical requirements.

Kind regards,

Harald Mischak

Academic Editor

PLOS ONE
---

## [Editor Report · Acceptance letter]

10 Nov 2020

PONE-D-20-24273R2 

Plasma and urine metabolomic analyses in aortic valve stenosis reveal shared and biofluid-specific changes in metabolite levels 

Dear Dr. Gauguier:

I'm pleased to inform you that your manuscript has been deemed suitable for publication in PLOS ONE. Congratulations! Your manuscript is now with our production department. 

Kind regards, 

on behalf of

Prof. Harald Mischak 

Academic Editor

PLOS ONE